# Unsupervised Model-based Pre-training for Data-efficient Control from Pixels

## Abstract

Controlling artificial agents from visual sensory data is an arduous task. Reinforcement learning (RL) algorithms can succeed in this but require large amounts of interactions between the agent and the environment. To alleviate the issue, unsupervised RL proposes to employ self-supervised interaction and learning, for adapting faster to future tasks. Yet, whether current unsupervised strategies improve generalization capabilities is still unclear, especially in visual control settings. In this work, we design an unsupervised RL strategy for data-efficient visual control. First, we show that world models pre-trained with data collected using unsupervised RL can facilitate adaptation for future tasks. Then, we analyze several design choices to adapt faster, effectively reusing the agents' pre-trained components, and planning in imagination, with our hybrid planner, which we dub Dyna-MPC. By combining the findings of a large-scale empirical study, we establish an approach that strongly improves performance on the Unsupervised RL Benchmark, requiring $20\times$ less data to match the performance of supervised methods. The approach also demonstrates robust performance on the Real-Word RL benchmark, hinting that the approach generalizes to noisy environments.

## 1 Introduction

Modern successes of deep reinforcement learning (RL) have shown promising results for control problems (Levine et al., 2016; OpenAI et al., 2019; Lu et al., 2021). However, training an agent for each task individually requires a large amount of task-specific environment interactions, incurring huge redundancy and prolonged human supervision. Developing algorithms that can efficiently adapt and generalize to new tasks has hence become an active area of research in the RL community.

In computer vision and natural language processing, unsupervised learning has enabled training models without supervision to reduce sample complexity on downstream tasks (Chen et al., 2020; Radford et al., 2019). In a similar fashion, unsupervised RL (URL) agents aim to learn about the environment without the need for external reward functions, driven by intrinsic motivation (Pathak et al., 2017; Burda et al., 2019a; Bellemare et al., 2016). Any learned models can then be adapted to downstream tasks, aiming to reduce the required amount of interactions with the environment.

Recently, the Unsupervised RL Benchmark (URLB) (Laskin et al., 2021) established a common protocol to compare self-supervised algorithms across several domains and tasks from the DMC Suite (Tassa et al., 2018). In the benchmark, an agent is allowed a task-agnostic pre-training stage, where it can interact with the environment in an unsupervised manner, followed by a fine-tuning stage where, given a limited budget of interactions with the environment, the agent should quickly adapt for a specific task. However, the results obtained by Laskin et al. (2021) suggest that current URL approaches may be insufficient to perform well on the benchmark, especially when the inputs of the agent are pixel-based images.

World models have proven highly effective for solving RL tasks from vision both in simulation (Hafner et al., 2021; 2019a) and in robotics (Wu et al., 2022), and they are generally data-efficient as they enable learning behavior in imagination (Sutton, 1991). Inspired by previous work on exploration (Sekar et al., 2020), we hypothesize this feature could be key in the unsupervised RL setting, as a pre-trained world model can leverage previous experience to learn behavior for new tasks in imagination, and in our work, we study how to best exploit this feature. We adopt the URLB setup to perform a large-scale study, involving several unsupervised RL methods for pre-training

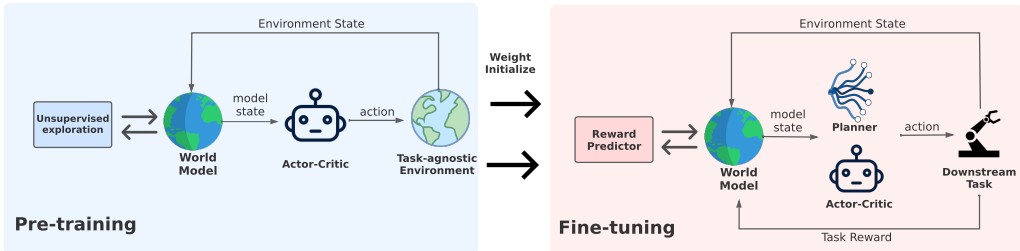

Figure 1: **Method overview.** Our method considers a pre-training (PT) and a fine-tuning (FT) stage. During pre-training, the agent interacts with the environment through unsupervised RL, maximizing an intrinsic reward function, and concurrently training a world model on the data collected. During fine-tuning, the agent exploits its pre-trained components and plans in imagination, to efficiently adapt to different downstream tasks, maximizing the rewards received from the environment.

model-based agents, different fine-tuning strategies, and a new improved algorithm for efficiently planning with world models. The resulting approach, which combines the findings of our study, strongly improves performance on the URL benchmark from pixels, nearly achieving the asymptotic performance of supervised RL agents, trained with 20x more task-specific data, and bridging the gap with low-dimensional state inputs (Laskin et al., 2021).

**Contributions.** This work does not propose a novel complex method. Rather, we study the interplay of various existing components and propose a novel final solution that outperforms existing state of the art on URLB by a staggering margin. Specifically:

- we demonstrate that unsupervised RL combined with world models can be an effective pre-training strategy to enable data-efficient visual control (Section 3.1),
- we study the interplays between the agent's pre-trained components that improve sample efficiency during fine-tuning (Section 3.2),
- we propose a novel hybrid planner we call *Dyna-MPC*, which allows us to effectively combine behaviors learned in imagination with planning (Section 3.3),
- combining our findings into one approach, we outperform previous approaches on URLB from pixels, nearly solving the benchmark (Section 4.1),
- we show the approach is resilient to environment perturbations, evaluating it on the Real World RL benchmark (Dulac-Arnold et al., 2020) (Section 4.2),
- we present an extensive analysis of the pre-trained agents, aimed at understanding in-depth the current findings and limitations (Section 4.3).

An extensive empirical evaluation, supported by more than 2k experiments, among main results, analysis and ablations, was used to carefully design our method. We hope that our large-scale evaluation will inform future research towards developing and deploying pre-trained agents that can be adapted with considerably less data to more complex/realistic tasks, as it has happened with unsupervised pre-trained models for vision (Parisi et al., 2022) and language (Ahn et al., 2022). [1]

## 2 PRELIMINARIES

**Reinforcement learning.** The RL setting can be formalized as a Markov Decision Process (MDP), denoted with the tuple $\{\mathcal{S}, \mathcal{A}, T, R, \gamma\}$, where $\mathcal{S}$ is the set of states, $\mathcal{A}$ is the set of actions, $T$ is the state transition dynamics, $R$ is the reward function, and $\gamma$ is a discount factor. The objective of an RL agent is to maximize the expected discounted sum of rewards over time for a given task, also called return, and indicated as $G_t = \sum_{k=t+1}^{T} \gamma^{(k-t-1)} r_k$. In continuous-action settings, you can learn an actor, i.e. a model predicting the action to take from a certain state, and a critic, i.e. a model that estimates the expected value of the actor's actions over time. Actor-critic algorithms can be combined with the expressiveness of neural network models to solve complex continuous control tasks (Haarnoja et al., 2018; Lillicrap et al., 2016; Schulman et al., 2017).

---

[1]The PyTorch code for the experiments will be open-sourced upon publication.

**Unsupervised RL.** In this work, we investigate the problem of fast adaptation for a downstream task, after a phase of unsupervised training and interaction with the environment. Our training routine, based on the setup of URLB (Laskin et al., 2021), is made of two phases: a pre-training (PT) phase, where the agent can interact with a task-agnostic version of the environment for up to 2M frames, and a fine-tuning phase (FT), where the agent is given a task to solve and a limited budget of 100k frames. During the PT phase, rewards are removed so that sensible information about the environment should be obtained by exploring the domain-dependent dynamics, which is expected to remain similar or unchanged in the downstream tasks. During FT, the agent receives task-specific rewards when interacting with the environment. As the agent has no prior knowledge of the task, it should both understand the task and solve it efficiently, in a limited interaction budget. In this setting, the performance of unsupervised model-free RL (Yarats et al., 2022) were shown to be insufficient as reported in (Laskin et al., 2021). We believe the key reason for this is that model-free RL algorithms can exploit only a little part of the information obtained with self-supervised interaction, as they rely uniquely on actor and critic's predictions.

**World models.** In this work, we ground upon the DreamerV2 agent (Hafner et al., 2021), which learns a world model (Ha & Schmidhuber, 2018; Hafner et al., 2019b) predicting the outcomes of actions in the environment. The dynamics is captured into a latent space $\mathcal{Z}$, providing a compact representation of the high-dimensional inputs. The world model consists of the following components:

$$
\begin{array}{llll}
\text{Encoder:} & e_t = f_\phi(s_t), & \text{Decoder:} & p_\phi(s_t|z_t), \\
\text{Dynamics:} & p_\phi(z_t|z_{t-1}, a_{t-1}), & \text{Posterior:} & q_\phi(z_t|z_{t-1}, a_{t-1}, e_t).
\end{array}
$$

The model states $z_t$ have both a deterministic component, modeled using the recurrent state of a GRU (Chung et al., 2014), and a (discrete) stochastic component. The encoder and decoder are convolutional neural networks (CNNs) and the remaining components are multi-layer perceptrons (MLPs). The world model is trained end-to-end by optimizing an evidence lower bound (ELBO) on the log-likelihood of the data collected in the environment (Hafner et al., 2019b;a). For the encoder and the decoder networks, we used the same architecture as in Hafner et al. (2021).

For control, the agent learns latent actor $\pi_\theta(a_t|z_t)$ and critic $v_\psi(z_t)$ networks. Both components are trained online within the world model, by imagining the model state outcomes of the actions produced by the actor, using the model dynamics. Rewards for imagined trajectories are provided by a reward predictor, $p_\phi(r_t|z_t)$ trained to predict environment rewards, and they are combined with the critic predictions to produce a GAE-$\lambda$ estimate of the returns (Schulman et al., 2016). The actor maximizes estimates of returns, backpropagating gradients through the model dynamics. The hyperparameters for the agent, which we keep fixed across all domains/tasks, can be found in Appendix H.

## 3 UNSUPERVISED MODEL-BASED PRE-TRAINING FOR DATA-EFFICIENT CONTROL FROM PIXELS

To best exploit self-supervised pre-training for data-efficient adaptation, it is important that the agent: *(i)* meaningfully interacts with the environment during the PT phase, to discover useful transitions; *(ii)* successfully reuses the modules learned during PT for fast adaptation; and *(iii)* efficiently employs the FT phase to quickly understand and master the downstream task. In this section, we use an experiment-driven approach to find which methods or components are best at tackling these challenges.

**Experimental procedure.** We employ the URL benchmark that consists of three control domains, Walker, Quadruped and Jaco, and twelve tasks, four per domain. To evaluate the agents, we take snapshots of the agent at different times during training, i.e. 100k, 500k, 1M, and 2M frames, and fine-tune the agent for 100k frames. In all bar plots, we show average normalized returns on downstream tasks with error bars showing the standard deviation. To normalize results in a comparable way for all tasks, we train a fully-supervised agent with 2M frames per task. We use the mean performance of this agent, which we refer to as "oracle", as the reference scores to normalize our results in the plots (details in Appendix A). For all experiments, results are presented with at least three random seeds.

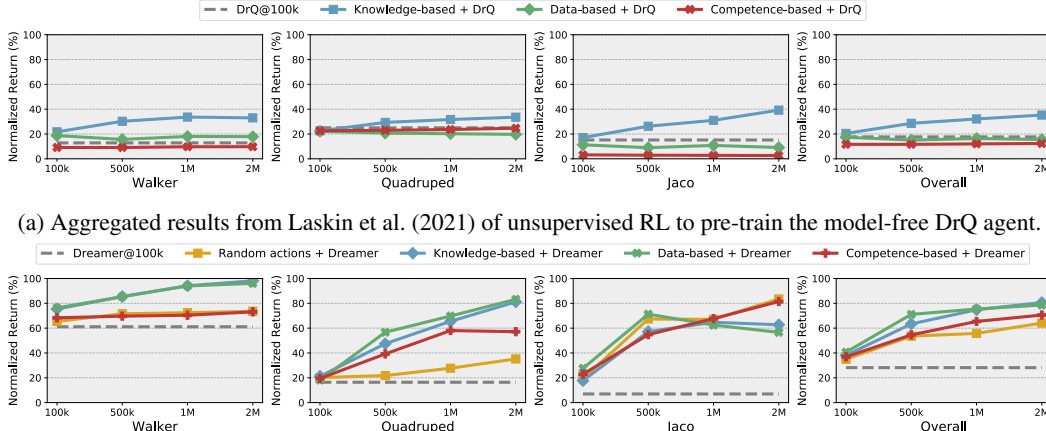

(a) Aggregated results from Laskin et al. (2021) of unsupervised RL to pre-train the model-free DrQ agent.

(b) Our aggregated results of unsupervised RL to pre-train the model-based Dreamer agent.

Figure 2: **Unsupervised pre-training.** Aggregated performance of different URL techniques for PT, with FT snapshots taken at different times along training. (a) With the model-free DrQ agent, performance slightly improves over time only using knowledge-based techniques. (b) With the model-based Dreamer agent, performance is higher and, overall, improves for all techniques. We also report Dreamer@100k and DrQ@100k results, which are obtained in 100k FT steps with no PT.

### 3.1 UNSUPERVISED PRE-TRAINING

In the PT stage, unsupervised RL can be used to explore the environment, collecting the data to train the components of the agent. The resulting networks are then used to initialize respective components in the agent deployed for the downstream task, aiming to reduce sample complexity during FT. The first question we address is thus "*What kinds of agents work best with unsupervised pre-training?*".

Unsupervised RL methods can be grouped into three categories (Laskin et al., 2021): knowledge-based, which aim to increase the agent's knowledge by maximizing error prediction (Pathak et al., 2017; 2019; Burda et al., 2019b), data-based, which aim to achieve diversity of data (Yarats et al., 2021; Liu & Abbeel, 2021b) and competence-based, which aim to learn diverse skills (Liu & Abbeel, 2021a; Eysenbach et al., 2019). In Figure 2a we report the results from Laskin et al. (2021), showing that none of these approaches is particularly effective on URLB when combined with the DrQ model-free agent (Yarats et al., 2022), state-of-the-art in RL from pixels, where the data collected with unsupervised RL is used to pre-train the agent's actor, critic, and encoder.

To demonstrate that world models can be used to effectively exploit unsupervised RL data collection for fast adaptation, we study multiple approaches and use them to pre-train the Dreamer's world model and latent actor. As knowledge-based methods we employ ICM (Pathak et al., 2017), LBS (Mazzaglia et al., 2021b), Plan2Explore (P2E; (Sekar et al., 2020)), and RND (Burda et al., 2019b). As a data-based approach, we choose APT (Liu & Abbeel, 2021b), and as competence-based approaches, we adopt DIAYN (Eysenbach et al., 2019) and APS (Liu & Abbeel, 2021a). Finally, we also test random actions, as a naive maximum entropy baseline (Haarnoja et al., 2018). Details on these methods and how we combined them with the Dreamer algorithm are discussed in Appendix B.

Aggregating results per category, in Figure 2b, we show that by leveraging a pre-trained world model the overall performance improves over time for all categories, as opposed to the model-free results, where only knowledge-based approaches slightly improve. In particular, data-based and knowledge-based methods are more effective in the Walker and Quadruped domains, and random actions and competence-based are more effective in the Jaco domain. Detailed results for each method are available in Appendix E.

### 3.2 FINETUNING PRE-TRAINED AGENTS

Some of the components learned during the PT phase, such as the world model, can be reused for fast adaptation during FT. However, as the reward is changing from pseudo-reward to task reward when

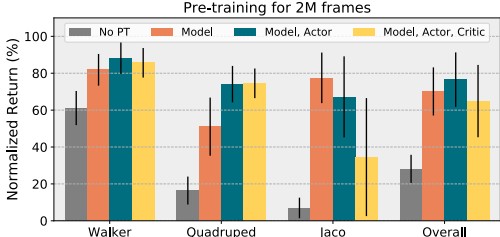 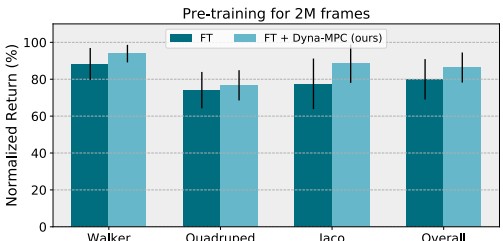

Figure 3: **Fine-tuning pre-trained agents.** Comparison of the results when fine-tuning different pre-trained components of the agent. Results averaged across all URL methods.

Figure 4: **Learning and planning.** Using Dyna-MPC during fine-tuning improves performance, increasing mean scores and reducing variance. Results averaged across all URL methods.

changing from the PT to the FT phase, it is not clear if pre-training of the actor and critic can help the downstream task. To shed light on this, we seek to answer: "*Which pre-trained components are useful for downstream tasks?*".

Here, we test different fine-tuning configurations, where we copy the weights of some of the PT components into the agent to fine-tune for the downstream task. We run the tests for the several unsupervised RL methods combined with Dreamer that we presented in Section 3.1 and show aggregated results in Figure 3 (detailed results per each method in Appendix E).

Overall, fine-tuning the PT world model provides the most significant boost in performance, strengthening the hypothesis that world models are very effective with unsupervised RL. Fine-tuning the actor improves performance slightly in Walker and remarkably in Quadruped, but is harmful in the Jaco domain. An intuitive explanation is that in the Quadruped and Walker moving tasks, the exploratory behaviors help discovering reward faster. Instead, in the Jaco goal-reaching tasks, the agent needs to reach a certain target with sparse rewards. If the PT actor is initialized to move far from the target, the agent might struggle to find rewards in the small FT budget. Finally, using a PT critic is systematically worse. This can be explained by the discrepancy between intrinsic rewards and task rewards.

### 3.3 Learning and Planning in Imagination

Knowing a model of the environment, traditional model-based control approaches, e.g. model predictive control (MPC) (Williams et al., 2015; Chua et al., 2018; Richards, 2005), can be used to plan the agent's action. Nonetheless, using actor-critic methods has several advantages, such as amortizing the cost of planning by caching previously computed (sub)optimal actions and computing long-term returns from a certain state, without having to predict outcomes that are far in the future. More recent hybrid strategies, such as LOOP (Sikchi et al., 2020) and TD-MPC (Hansen et al., 2022), allow combining trajectories sampled from the actor with trajectories sampled from a distribution over actions that is iteratively improved. The model and the critic are used to evaluate the trajectories,

---

**Algorithm 1** Dyna-MPC

**Require:** Actor $\theta$, Critic $\psi$, World Model $\phi$
1:      $\mu, \sigma$: initial parameters for sampling actions
2:      $N, N_\pi$: num trajectories, num policy trajectories
3:      $\mathbf{z}_t, H$: current model state, planning horizon
4: **for** each iteration $j = 1..J$ **do**
5:      Sample $N$ trajectories of length $H$ from $\mathcal{N}(\mu, \sigma^2 \mathrm{I})$, starting from $z_t$
6:      Sample $N_\pi$ trajectories of length $H$ using the actor $\pi_\theta$, starting from $z_t$
7:      Estimate future states, using the model, and returns, using reward and critic predictions
8:      Update $\mu$ and $\sigma$ using MPPI (Williams et al., 2015)
9: **end for**
10: **return** $\mathbf{a_t} \sim \mathcal{N}(\mu_t, \sigma_t^2 \mathrm{I})$

---

improve them, and eventually select the most promising actions, i.e. planning. In this section, we answer the question: *Can we accelerate downstream task adaptation by leveraging planning?*

**Dyna-MPC.** As we pre-train a world model, we could exploit planning in latent space to adapt with limited additional environment interaction. One problem with the above strategies is that they are based upon learning off-policy actor and critic, which in our context would prevent us from exploiting the PT model to learn the actor and critic in imagination. In order to enable hybrid planning with the behavior learned in imagination (Hafner et al., 2019a), we develop a modification of these approaches, which we call *Dyna-MPC*, that combines the actor and critic learned in imagination with MPPI (Williams et al., 2015) for planning.

As detailed in Algorithm 1, at each time step, we imagine a set of latent trajectories using the model, by sampling actions from a time-dependent multivariate gaussian and from the actor policy, trained with Dreamer in imagination. Returns for MPPI are estimated using reward predictions by the model and the critic. MPPI is used to update the parameters of the multivariate gaussian for $J$ iterations. Details on how returns are estimated and the MPPI updates work are given in Appendix C. One significant difference with previous approaches is that the policy in Dyna-MPC is learned on-policy in imagination, thus no correction for learning off-policy is required (Sikchi et al., 2020).

Given the insights from the previous section, we use the world models and actors pre-trained with all the different unsupervised strategies we considered (see Section 3.1)[2] and test their FT performance with and without planning with Dyna-MPC. Aggregated scores are reported in Figure 4, and detailed results for each method are available in Appendix E. We observe that adopting Dyna-MPC is always beneficial, as it improves the average performance and reduces variance in all domains.

## 3.4 OUR METHOD: COMBINING THE FINDINGS TOGETHER

In the large-scale study, we explored several design choices to establish the most adequate approach to tackle the URL benchmark, aiming to provide a general recipe for data-efficient adaptation thanks to unsupervised RL. Our approach combines the main findings we presented in the previous sections:

1. learning a model-based agent with data collected using unsupervised RL (Figure 2);
2. fine-tuning the PT world model (always) and the pre-trained actor (where beneficial), while learning the critic from scratch (Figure 3);
3. adopting a hybrid planner, as the proposed Dyna-MPC, to leverage both learning and planning in imagination (Figure 4).

An overview of the method is illustrated in Figure 1 and the algorithm is presented in Appendix D. We believe the above recipe could be generally applied to unsupervised settings, also outside of URLB, with the precaution that one should carefully make two decisions: (a) whether fine-tuning the PT actor is meaningful for the downstream task or it's better to re-learn it from scratch, (b) what is the best URL strategy to collect data. Both decisions strongly depend on the target domain/task and so it is difficult to assess their implications beforehand. However, adopting unsupervised strategies that specifically focus on interacting with interesting elements of the environment, e.g. objects, or that quickly explore large areas of the environment at the beginning of fine-tuning may help exploring and revisiting crucial states of the environment more easily (Parisi et al., 2021).

For URLB, we already established (a) that the PT actor is effective in Walker and Quadruped tasks, but it is better re-learn the actor from scratch in Jaco, in Section 3.2. To decide which URL strategy to use (b) we present a detailed comparison of the performance of our approach using different exploration strategies. The results in Figure 5 show that the agent using LBS during pre-training performs overall best, as it has the highest interquartile mean (IQM) and mean scores, and the lowest optimality gap. Thus, in the evaluation section, we present **Ours (LBS)** as our approach.

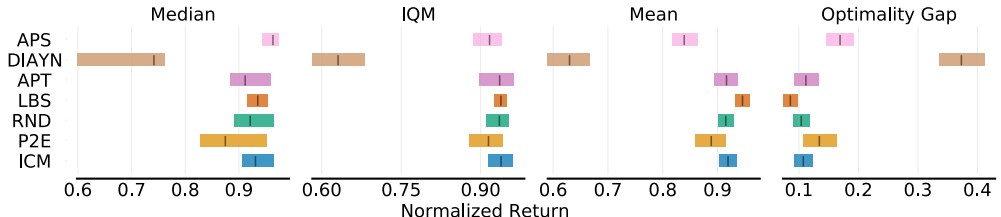

Figure 5: **Unsupervised RL comparison.** Stratified bootstrapped performance of our approach on URLB under different data collection strategies, with 95% CIs. (Agarwal et al., 2021).

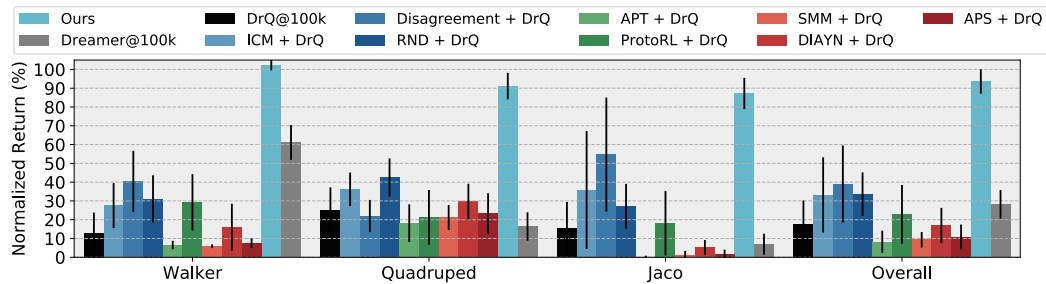

Figure 6: **URLB results.** We compare our approach to the results on URLB after 2M frames of pre-training from (Laskin et al., 2021) (DrQ pre-trained with different unsupervised RL strategies).

## 4 EVALUATION AND ANALYSIS

### 4.1 UNSUPERVISED REINFORCEMENT LEARNING BENCHMARK

In Section 3, we presented our approach, which combines the findings from our empirical large-scale study on URLB. In Figure 6, we compare the results from the original URLB paper with our approach. The performance of our method is superior in all domains. The second strongest method (DrQ with Disagreement) approaches an overall performance of 40% of the respective supervised baseline performance, while our method recovers more than 90% of its supervised counterpart.

### 4.2 REAL-WORLD REINFORCEMENT LEARNING BENCHMARK

Algorithms developed in simulation struggle to transfer to real-world systems due to a series of implicit assumptions that are rarely satisfied in real environments, e.g. URLB assumes the dynamics between PT and FT stay the same. The RWRL benchmark (Dulac-Arnold et al., 2020) considers several challenges that are common in real-world systems and implements them on top of DMC tasks.

We employ vision-based variants of the Walker Walk and Quadruped Walk tasks from the RWRL benchmark. These tasks introduce system delays, stochasticity, and perturbations of the robot's model and sensors, which are applied with three degrees of intensity to the original environment, i.e. 'easy', 'medium', and 'hard' (details in Appendix F). We seek to answer whether in perturbed settings:

- does unsupervised PT enable faster adaptation?
- does unsupervised RL provide an advantage over random exploration?
- does hybrid planning improve performance, as in URLB?

In Figure 7, we present the results of our method, using LBS during PT, with and without planning with Dyna-MPC for FT, and compare to random exploration and training from scratch for 100k, 1M, and 2M frames. Crucially, the PT models are trained in the vanilla task-agnostic version of the environments from the DMC Suite, so that the results highlight the extent to which models trained in ideal conditions generalize to perturbed settings when fine-tuned in a low-data regime.

---

[2]We exceptionally do not use the pre-trained actor in the Jaco tasks, as this was shown to lead to better performance in Section 3.2 (Figure 3).

Overall, we found that fine-tuning PT models offer an advantage over training from scratch for 100k frames, despite all the variations in the environment. Furthermore, on the Quadruped Easy and Medium settings, our method performs better than Dreamer@1M and not far from Dreamer@2M while using 10x and 20x less task-specific data, respectively. Our method also performs close to Dreamer@1M/2M in the Walker Easy task. Unsupervised RL for data collection (Ours) outperforms random actions in the 'easy' and 'medium' settings, showing that a better PT model yields higher FT performance, even when the dynamics of the downstream task is affected by misspecifications and noisy factors. Finally, in contrast with the findings on URLB, adopting the hybrid planner is not generally beneficial. We believe this is because the model's predictions are less certain and precise in this setting and thus cannot inform the short-term planner accurately.

## 4.3 EXTENDED ANALYSIS

To better analyze the learned components, we conducted a range of additional experiments. For conciseness, detailed descriptions of the experimental settings are deferred to Appendix G and we briefly summarize the takeaways in this section.

**Learning rewards online.**   We verify whether having to discover and learn the reward function during FT impacts performance. In Figure 8, we compare against agents that (violating the URLB settings) know the task in advance and can pre-train a reward predictor during the PT stage. We see that learning the reward predictor does not affect performance significantly for dense-reward tasks, such as the Walker and Quadruped tasks. However, in sparser reward tasks, i.e. the Jaco ones, knowing reward information in advance provides an advantage. Efficient strategies to find sparse rewards efficiently represent a challenge for future research. More details in Appendix G.1.

**Zero-shot adaptation.**   Knowing a reward predictor from PT, it could be possible to perform zero-shot control with MPC methods if the model and the reward function allow it. In Figure 9, we show that despite the zero-shot MPC (ZS) offers an advantage over Dreamer@100k, the FT phase is crucial to deliver high performance on the downstream tasks, as the agent uses this phase to collect missing information about the environment and the task. Further details in Appendix G.2.

**Latent dynamics discrepancy (LDD).**   We propose a novel metric, *Latent Dynamics Discrepancy*, which evaluates the distance between the latent predictions of the PT model and the same model after FT on a task. In Figure 10, we show the correlation between our metric and the performance ratio between using the PT model and the FT model for planning (see Appendix G.3 for a detailed explanation). We observed a strong negative Pearson correlation ($-0.62$, p-value: $0.03$), highlighting that major updates in the model dynamics during FT played an important role in improving performance.

**Unsupervised rewards and performance.** We analyze the correlation between the normalized performance of different agents and their intrinsic rewards for optimal trajectories obtained by an oracle agent in Table 1. In particular, the correlation for LBS, which overall performs best in URLB, has a statistical significance, as its p-value is $< 0.05$. We believe this correlation might be one of the causes of LBS outstanding performance. Further insights are provided in Appendix G.4.

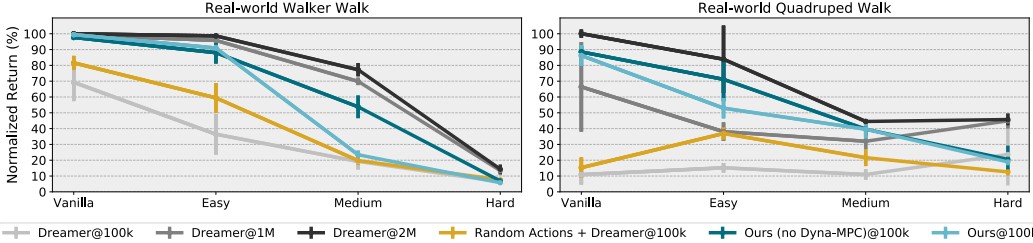

Figure 7: **RWRL results.** We compare our method to random exploration and training from scratch on the tasks from the RWRL benchmark. Our models are pre-trained on the vanilla version of the environment for 2M frames and fine-tuned for 100k frames on the perturbated tasks from RWRL.

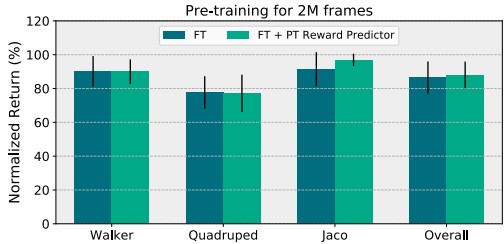
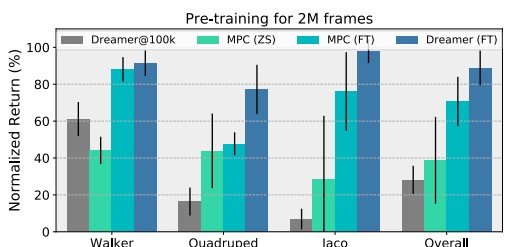

Figure 8: **Reward learning ablation.** Results averaged across all unsupervised RL methods.

Figure 9: **Zero-shot (ZS) vs Fine-tuned (FT).** Results of the agent using Plan2Explore.

## 5 RELATED WORK

**Model-based control.** Dynamics models combined with powerful search methods have led to impressive results on a wide variety of tasks such as Atari (Schrittwieser et al., 2020) and continuous control (Hafner et al., 2019a; Janner et al., 2019; Sikchi et al., 2021; Lowrey et al., 2018). LOOP (Sikchi et al., 2020) and TD-MPC (Hansen et al., 2022) combine temporal difference learning and MPC. The model proposed with TD-MPC is task-oriented and thus requires a task to accelerate learning. In our work, we focus on unsupervised model learning, grounding on the DreamerV2 model (Hafner et al., 2021), whose supervision comes from predicting the environment's observations. Methods that use no reconstruction could generalize better to visual differences (Mazzaglia et al., 2021a; Ma et al., 2020) but they lose in explainability, as they cannot decode imagined trajectories.

**Unsupervised RL.** Prior to our work, the large-scale study of curiosity (Burda et al., 2018) provided an insightful analysis of the performance of knowledge-based methods in the reward-free setting. In our work, we leverage the URLB setting, to provide an analysis of a combination of model-based control techniques with unsupervised RL. This allowed us to formulate a strategy to adapt pre-trained models to visual control tasks in a data-efficient manner. Closely, Sekar et al. (2020) combines adapts the Disagreement (Pathak et al., 2019) to work with Dreamer (Hafner et al., 2019a). In our work, in addition to analyzing a wider choice of unsupervised RL strategies, we show how to better exploit the agent PT components for adaptation, and we propose a hybrid planner to improve data-efficiency.

**Transfer learning.** In the field of transfer learning, fine-tuning is the most used approach. However, fine-tuning all the pre-trained agent components may not be the most effective strategy. In transfer learning for RL, they have studied this problem, mainly with the objective of transferring from one environment to another (Farebrother et al., 2018; Sasso et al., 2022; van Driessel & Francois-Lavet, 2021). Instead, we analyze which agent's components should be transferred from the unsupervised PT stage to the supervised FT stage when the environment's dynamics is assumed to stay similar or be the same. Another stream of work has studied successor representations, to enable a better transfer of the agent's actor-critic (Hansen et al., 2020; Barreto et al., 2016).

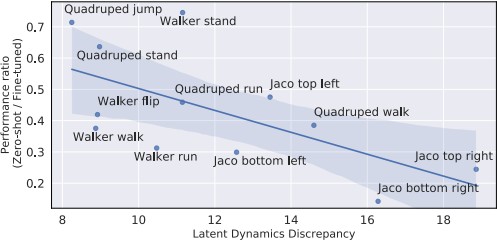

Figure 10: **LDD and performance correlation.** The line and shaded area represent a linear regression model fit and its confidence intervals.

Table 1: **Pearson correlation and p-value** between fine-tuned performance across URLB tasks and intrinsic rewards.

| Pre-training for 2M environment frames | | | | |
|---|---|---|---|---|
| | ICM | LBS | P2E | RND |
| Correlation | -0.54 | -0.60 | -0.34 | -0.03 |
| p-value | 0.07 | **0.04** | 0.28 | 0.91 |

## 6 CONCLUSION

In order to accelerate the development and deployment of learning agents for real-world tasks, it is crucial that the employed algorithms can adapt in a data-efficient way for multiple tasks. Our study provides an empirical analysis of several design choices, which allowed us to obtain near-optimal performance in URLB and that showed robustness to perturbations in the environment, on the RWRL benchmark. We also analyzed several aspects of the learned models, to understand what could be improved further in the future to ease the adaptation process.

**Limitations.** In the Jaco reaching tasks, we found that a bad initialization of the pre-trained actor can actually harm the agent's performance. While competence-based approaches should address this limitation, by learning a variety of skill behaviors, their performance on the other domains has been subpar. Future work should aim to find a more general approach to pre-train behavior for fast adaptation or improve the exploration capabilities of competence-based approaches.

Another issue we encountered, on the RWRL benchmark, is that if the environment introduces too intense perturbations during adaptation, relying on the predictions of the adopted world model becomes problematic, to the extent that exploiting a planner is not useful anymore. Developing more resilient models that can be trained in an unsupervised fashion and used for data-efficient planning, even in presence of complex perturbations, will be the focus of future studies.

**Reproducibility statement** We reported in the main text (Algorithm 1) the pseudo-code for Dyna-MPC and in Appendix D the pseudo-code for our end-to-end approach. We also provide instructions on how we implemented our methods (Appendix B) and all the model and training hyperparameters to implement and reproduce the results (Table 4). We will release our code and scripts.

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

APPENDIX

## A    NORMALIZATION SCORES

| Pre-trainining for 2M environment frames | | | | | |
|---|---|---|---|---|---|
| Domain | Task | URLB Expert | URLB Disagreement | Dreamer@2M | Ours |
| Walker | Flip | 799 | $346 \pm 41$ | 778 | $938 \pm 12$ |
| | Run | 796 | $208 \pm 47$ | 724 | $596 \pm 38$ |
| | Stand | 984 | $746 \pm 107$ | 909 | $973 \pm 14$ |
| | Walk | 971 | $549 \pm 117$ | 965 | $959 \pm 1$ |
| Quadruped | Jump | 888 | $389 \pm 196$ | 753 | $822 \pm 33$ |
| | Run | 888 | $337 \pm 95$ | 904 | $642 \pm 99$ |
| | Stand | 920 | $512 \pm 281$ | 945 | $927 \pm 28$ |
| | Walk | 866 | $293 \pm 117$ | 947 | $816 \pm 61$ |
| Jaco | Reach bottom left | 193 | $124 \pm 22$ | 223 | $192 \pm 23$ |
| | Reach bottom right | 203 | $115 \pm 32$ | 231 | $192 \pm 19$ |
| | Reach top left | 191 | $106 \pm 38$ | 233 | $197 \pm 17$ |
| | Reach top right | 223 | $139 \pm 22$ | 225 | $212 \pm 13$ |

Table 2: Performance of expert baseline and the best method on pixel-based URLB from Laskin et al. (2021) and performance of our oracle baseline (Dreamer@2M) and best approach, using LBS for unsupervised data collection, after pre-training for 2M frames and fine-tuning for 100k steps.

In Table 2, we report the mean scores for the URLB Expert, used to normalize the scores in the URLB paper, and for Dreamer@2M, which we use to normalize returns of our methods, where both supervised baselines have been trained individually on each of the 12 tasks from URLB for 2M frames. We additionally report mean and standard deviations for the best performing unsupervised baseline from URLB. which is Disagreement (Pathak et al., 2019), and our method (using LBS for data collection). We notice that our scores approach the Dreamer@2M's scores in several tasks, eventually outperforming them in a few tasks (e.g. Walker Flip, Quadruped Jump). We believe this merit is due both to the exploration pre-training, which may have found more rewarding trajectories than greedy supervised RL optimization and of the improved Dyna-MPC planning strategy.

## B    INTEGRATING UNSUPERVISED RL STRATEGIES

We summarize here the unsupervised RL approaches tested and how we integrated them with the Dreamer algorithm for exploration. For all methods, rewards have been normalized during training using an exponential moving average with momentum $0.95$, with the exceptions of RND, which follows its original reward normalization (Burda et al., 2019b), and APS, whose rewards are not normalized because they are used to regress the skill that is closer to the downstream task during FT.

**ICM.**   The Intrinsic Curiosity Module (ICM; Pathak et al. (2017)) defines intrinsic rewards as the error between states projected in a feature space and a feature dynamics model's predictions. We use the Dreamer agent encoder $e_t = f_\phi(s_t)$ to obtain features and train a forward dynamics model $g(e_t|e_{t-1}, a_{t-1})$ to compute rewards as:

$$r_t{}^{\text{ICM}} \propto \|g(e_t|e_{t-1}, a_{t-1}) - e_t\|^2.$$

As the rewards for ICM require environment states (going through the encoder to compute prediction error), we train a reward predictor to allow estimating rewards in imagination.

**Plan2Explore.**   The Plan2Explore algorithm (Sekar et al., 2020) is an adaptation of the Disagreement algorithm (Pathak et al., 2019) for latent dynamics models. An ensemble of forward dynamics models is trained to predict the features embedding $e_t = f_\phi(s_t)$, given the previous latent state and actions, i.e. $g(e_t|z_{t-1}, a_{t-1}, w_k)$, where $w_k$ are the parameters of the k-th predictor. Intrinsic rewards are defined as the variance of the ensemble predictions:

$$r_t{}^{\text{P2E}} \propto \text{Var}(\{g(e_t|z_{t-1}, a_{t-1}, w_k)|k \in [1, ..., K]\}).$$

Plan2Explore requires only latent states and actions, thus it can be computed directly in imagination. We used an ensemble of 5 models.

**RND.** Random Network Distillation (RND; Burda et al. (2019b)) learns to predict the output of a randomly initialized network $n(s_t)$ that projects the states into a more compact random feature space. As the random network is not updated during training, the prediction error should diminish for already visited states. The intrinsic reward here is defined as:

$$r_t^{\text{RND}} \propto \|g(s_t) - n(s_t)\|^2$$

As the rewards for RND requires environment states (to encode with the random network), we train a reward predictor to allow estimating rewards in imagination.

**LBS.** In Latent Bayesian Surprise (LBS; Mazzaglia et al. (2021b)), they use the KL divergence between the posterior and the prior of a latent dynamics model as a proxy for the information gained with respect to the latent state variable, by observing new states. Rewards are computed as:

$$r_t^{\text{LBS}} \propto D_{\text{KL}}[q(z_t|z_{t-1}, a_{t-1}, e_t)\|p(z_t|z_{t-1}, a_{t-1})]$$

As the rewards for LBS requires environment states (to compute the posterior distribution), we train a reward predictor to allow estimating rewards in imagination.

**APT.** Active Pre-training (APT; Liu & Abbeel (2021b)) uses a particle-based estimator based on the K nearest-neighbors algorithm (Singh et al., 2003) to estimate entropy for a given state. We implement APT on top of the deterministic component of the latent states $\bar{z}_t$, providing rewards as:

$$r_t^{\text{APT}} \propto \sum_i^k \log \|\bar{z}_t - \bar{z}_t^i\|^2,$$

where $k$ are the nearest-neighbor states in latent space. As APT requires only latent states, it can be computed directly in imagination. We used $k = 12$ nearest neighbors.

**DIAYN.** Diversity is All you need (DIAYN; Eysenbach et al. (2019)) maximizes the mutual information between the states and latent skills $w$. We implement DIAYN on top of the latent space of Dreamer, writing the mutual information as $I(w_t, z_t) = H(w_t) - H(w_t|z_t)$. The entropy $H(w_t)$ is kept maximal by sampling $w_t \sim \text{Unif}(w_t)$ from a discrete uniform prior distribution, while $H(w_t|z_t)$ is estimated learning a discriminator $q(w_t|z_t)$. We compute intrinsic rewards as:

$$r_t^{\text{DIAYN}} \propto \log q(w_t|z_t)$$

Additionally, DIAYN maximizes the entropy of the actor, so we add an entropy maximization term to Dreamer's objective (Haarnoja et al., 2018). As DIAYN requires model states and skills sampled from a uniform distribution to compute rewards, we can directly compute them in imagination. For FT, the skill adapted is the one with the highest expected rewards, considering the states and rewards obtained in the initial episodes.

**APS.** Active Pre-training with Successor features (APS; Liu & Abbeel (2021a)) maximizes the mutual information between the states and latent skills $w$. We implement APS on top of the latent space of Dreamer, writing the mutual information as $I(w_t, z_t) = H(z_t) - H(z_t|w_t)$. The entropy term $H(z_t)$ is estimated using a particle-based estimator on top of the deterministic component of the latent states $\bar{z}_t$, as for APT, while the term $H(z_t|w_t)$ is estimated learning a discriminator $q(z_t|w_t)$. The intrinsic rewards for APS can be written as:

$$r_t^{\text{APS}} \propto r_t^{\text{APT}} + \log q(w_t|z_t)$$

As APS requires model states and uniformly sampled skills to compute rewards, we can directly compute them in imagination. For FT, the skill to adapt is selected using linear regression over the states and rewards obtained in the initial episodes (Liu & Abbeel, 2021a).

## C  DYNA-MPC

To further improve data efficiency, we chose to use an hybrid planner that combines reinforcement learning and MPC (Hansen et al., 2022; Sikchi et al., 2020; Lowrey et al., 2018). Previous works leveraged model-free off-policy algorithms (Hansen et al., 2022; Sikchi et al., 2020) to learn the actor and critic in a more computationally efficient manner. The policy used to act on the environment combines action samples from the actor network with MPC, while the critic and the actor are learned "offline" from previously collected data. This has several benefits but also leads to an issue referred to as "actor divergence" (Sikchi et al., 2020), which consists of the policy used for data collection being different from the policy that is used to learn the critic.

In our study, we found that using the PT world model to learn the actor and the critic is crucial to improve data-efficiency during FT (see Figure 3). Thus, we discard the option of learning the actor and critic with off-policy deep RL. Instead, we design a new hybrid planner, which we call Dyna-MPC, that learns actor and critic functions in the model imagination (Sutton, 1991), using the Dreamer algorithm (Hafner et al., 2019a), and then combines their predictions with MPPI (Williams et al., 2015) for acting on the environment. By doing so we mitigate the "actor divergence" issue as actor and critic are learned on-policy on the trajectories generated with the model.

The critic is learned in the model's imagination, computing the expected value of the actor's actions using GAE-$\lambda$ estimates of the returns (Schulman et al., 2016; Hafner et al., 2019a):

$$V_t^\lambda = r_t + \gamma_t \begin{cases} (1-\lambda)v_\psi(z_{t+1}) + \lambda V_{t+1}^\lambda & \text{if} \quad t < H, \\ v_\psi(z_H) & \text{if} \quad t = H, \end{cases} \tag{1}$$

where $r_t$ is the reward for state $z_t$, yielded by the reward predictor of the world model, and $H$ is the imagination horizon. When computing returns for MPPI we use the same return estimates.

At each time step, we use MPPI to select the best action. MPPI iteratively fits the parameters of a time-dependent multivariate Gaussian distribution with diagonal covariance, updating mean and standard deviation parameters using an importance weighted average of the top-k trajectories with the highest estimated returns. At every step, $N$ trajectories $\Gamma_i = \{a_{0,i}, a_{1,i}, ..., a_{H,i}\}$ of length $H$ are obtained sampling actions from the distributions $a_t \sim \mathcal{N}(\mu_t, \sigma_t^2 \mathrm{I})$ and $N_\pi$ trajectories are sampled from the actor network $a_t \sim \pi_\theta(a_t|z_t)$ and their outcomes are predicted using the model. At each MPPI iteration, the distribution parameters are updated as follows:

$$\mu = \frac{\sum_{i=1}^k \Omega_i \Gamma_i^\star}{\sum_{i=1}^N \Omega_i} \,, \ \sigma = \max \left( \sqrt{\frac{\sum_{i=1}^N \Omega_i (\Gamma_i^\star - \mu)^2}{\sum_{i=1}^N \Omega_i}} \,, \ \epsilon \right) \,, \tag{2}$$

where $\Omega_i = \exp(\tau V_i^\lambda)$, $\tau$ is a temperature parameter, $\star$ indicates the trajectory is in the top-k, and $\epsilon$ is a clipping factor to avoid too small standard deviations (Hansen et al., 2022). To reduce the number of iterations required for convergence, we reuse the 1-step shifted mean obtained at the previous timestep (Argenson & Dulac-Arnold, 2020).

## D ALGORITHM

---

**Algorithm 2** Unsupervised Model-based Pre-Training for Data-efficient Control from Pixels

---

**Require:** Actor $\theta$, Critic $\psi$, World Model $\phi$

  1:        Intrinsic reward $r^{\text{int}}$, extrinsic reward $r^{\text{ext}}$
  2:        Environment, $M$, downstream tasks $T_k$, $k \in [1, \ldots, M]$
  3:        Pre-train frames $N_{\text{PT}}$, fine-tune frames $N_{\text{FT}}$, environment frames/update $\tau$
  4:        Initial model state $z_0$, hybrid planner Dyna-MPC, replay buffers $\mathcal{D}_{\text{PT}}$, $\mathcal{D}_{\text{FT}}$
  5:
  6:  *// Pre-training*
  7:  **for** $t = 0, \ldots, N_{\text{PT}}$ **do**
  8:      Draw action from the actor, $\mathbf{a}_t \sim \pi_\theta(a_t | z_t)$
  9:      Apply action to the environment, $\mathbf{s}_{t+1} \sim P(\cdot | \mathbf{s}_t, \mathbf{a}_t)$
10:      Add transition to replay buffer, $\mathcal{D}_{\text{PT}} \leftarrow \mathcal{D}_{\text{PT}} \cup (\mathbf{s}_t, \mathbf{a}_t, \mathbf{s}_{t+1})$
11:      Infer model state, $z_{t+1} \sim q(z_{t+1} | z_t, a_t, f_\phi(s_{t+1}))$
12:      **if** $t \mod \tau = 0$ **then**
13:          Update world model parameters $\phi$ on the data from the replay buffer $\mathcal{D}_{\text{PT}}$
14:          Update actor-critic parameters $\{\theta, \psi\}$ in imagination, maximizing $r^{\text{int}}$
15:      **end if**
16:  **end for**
17:  Output pre-trained parameters $\{\psi_{\text{PT}}, \theta_{\text{PT}}, \phi_{\text{PT}}\}$
18:
19:  *// Fine-tuning*
20:  **for** $T_k \in [T_1, \ldots, T_M]$ **do**
21:      Initialize fine-tuning world-model with $\phi_{\text{PT}}$
22:      (Optional) Initialize fine-tuning actor with $\theta_{\text{PT}}$
23:      **for** $t = 0, \ldots, N_{\text{FT}}$ **do**
24:          Draw action from the actor, $\mathbf{a}_t \sim \pi_\theta(a_t | z_t)$
25:          Use the planner for selecting best action, $\mathbf{a}_t \sim \text{Dyna-MPC}(z_t)$
26:          Apply action to the environment, $\mathbf{s}_{t+1}, r_t^{\text{ext}} \sim P(\cdot | \mathbf{s}_t, \mathbf{a}_t)$
27:          Add transition to replay buffer, $\mathcal{D}_{\text{FT}} \leftarrow \mathcal{D}_{\text{FT}} \cup (\mathbf{s}_t, \mathbf{a}_t, r_t^{\text{ext}}, \mathbf{s}_{t+1})$
28:          Infer model state, $z_{t+1} \sim q(z_{t+1} | z_t, a_t, f_\phi(s_{t+1}))$
29:          **if** $t \mod \tau = 0$ **then**
30:             Update world model parameters $\phi$ on the data from the replay buffer $\mathcal{D}_{\text{FT}}$
31:             Update actor-critic parameters $\{\theta, \psi\}$ in imagination, maximizing $r^{\text{ext}}$
32:          **end if**
33:      **end for**
34:      Evaluate performance on $T_k$
35:  **end for**

---

# E   ADDITIONAL RESULTS

We present complete results, for each unsupervised RL method, for the large-scale study experiments presented in Section 3.

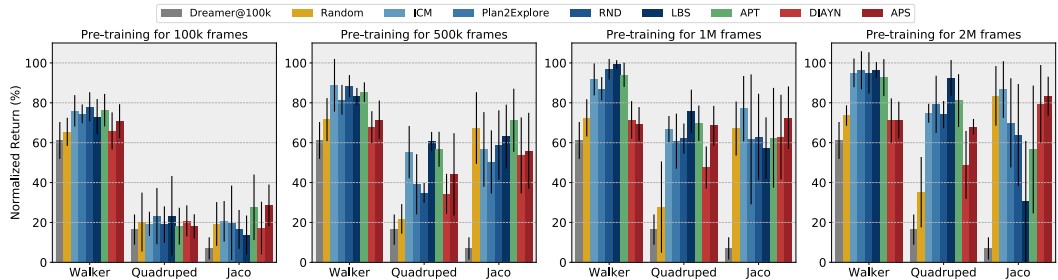

Figure 11: Complete results for Section 3.1 (Figure 2b).

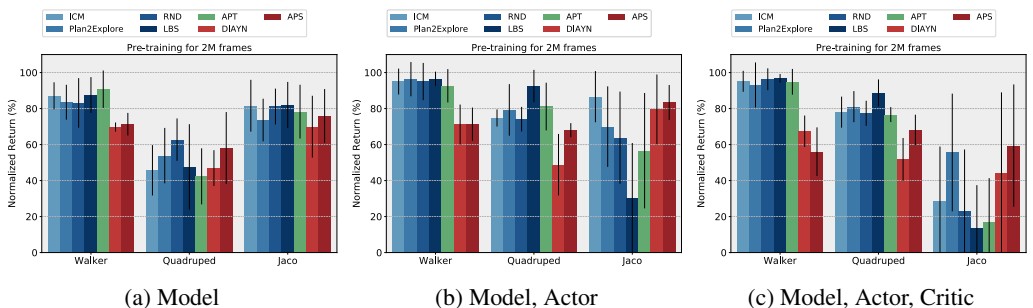

Figure 12: Complete results for Section 3.2 (Figure 3).

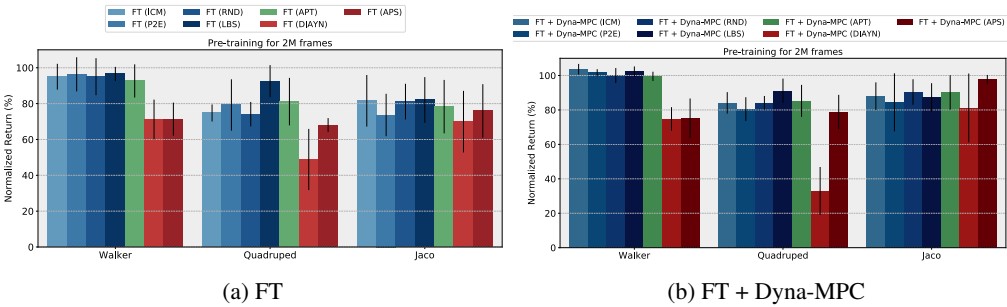

Figure 13: Complete results for Section 3.3 (Figure 4).

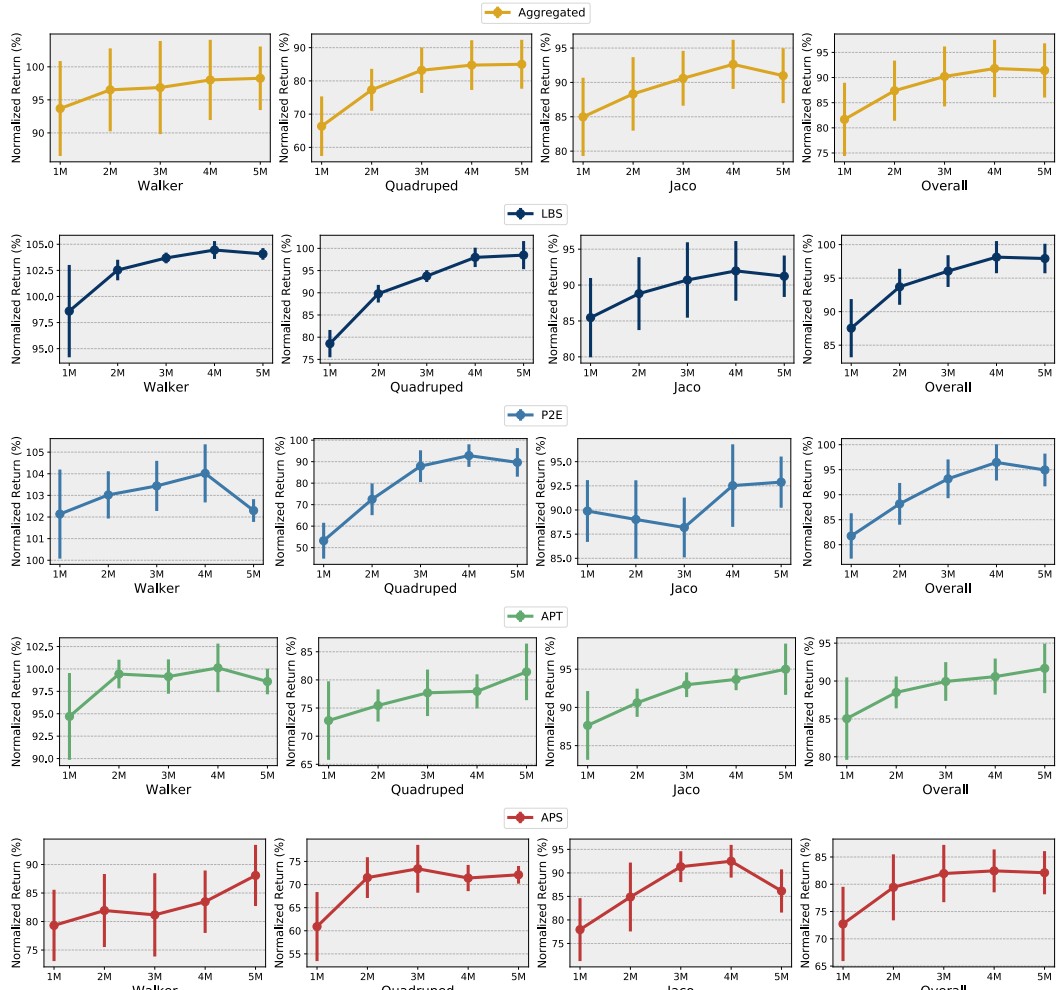

Figure 14: **Longer pre-training.** Fine-tuning performance of our method when pre-training for longer than 2M steps. Every bar reports mean performance and standard errors.

*Can a pre-training stage longer than 2M frames be beneficial?* In Figure 14, we report FT results with our full method, every 1M frames up to 5M PT frames. The aggregated results show that, adopting our method, longer PT can increase performance further, especially until 4M steps. The performance in all domains keeps increasing or remains steady until 5M steps, with two exceptional cases, Walker for Plan2Explore and Jaco for APS, where performance drops between 4M and 5M steps.

For these experiments, we kept the size of the model and all the hyperparameters unvaried with respect to the 2M PT frames experiments but we increased the replay buffer maximum size to 5M frames. Increasing model capacity, and adopting additional precautions, such as annealing learning rate, it is possible that the agent could benefit even more from longer pre-training and we aim to analyse this more in details for future work.

## F   RWRL SETTINGS

We take the Quadruped and Walker tasks from the RWRL benchmark and replace the low-dimensional sensor inputs with RGB camera inputs. While this removes some of the perturbations planned in the benchmark (Dulac-Arnold et al., 2020), such as noise in the sensors, it introduces the difficulty of a different dynamics in pixel space (due to the other perturbations), compared to the one observed during pre-training in the vanilla simulation environment.

| Setting | Easy | | Medium | | Hard | |
|---|---|---|---|---|---|---|
| **System Delays** | *Time Steps* | | *Time Steps* | | *Time Steps* | |
| Action | 3 | | 6 | | 9 | |
| Rewards | 10 | | 20 | | 40 | |
| **Action Repetition** | 1 | | 2 | | 3 | |
| **Gaussian Noise** | *Std. Deviation* | | *Std. Deviation* | | *Std. Deviation* | |
| Action | 0.1 | | 0.3 | | 1.0 | |
| **Perturbation Quadruped** | *[Min,Max]* | *Std.* | *[Min,Max]* | *Std.* | *[Min,Max]* | *Std.* |
| (shin length) | [0.25, 0.3] | 0.005 | [0.25, 0.8] | 0.05 | [0.25, 1.4] | 0.1 |
| **Perturbation Walker** | *[Min,Max]* | *Std.* | *[Min,Max]* | *Std.* | *[Min,Max]* | *Std.* |
| (thigh length) | [0.225, 0.25] | 0.002 | [0.225, 0.4] | 0.015 | [0.15, 0.55]] | 0.04 |

Table 3: Perturbations setting for each challenge of our adapted tasks from the RWRL benchmark, in increasing levels of intensity.

## G   EXTENDED ANALYSIS

We note that, to run the experiments faster, we did not use Dyna-MPC for the extended analysis. Furthermore, the Jaco tasks used slightly differ from the original ones in URLB, only in that the target to reach cannot move. This allows consistency of the reward function between PT and FT, so that a reward predictor can be trained on 'reward-labelled' PT data. However, because of this change, the performance in Jaco may differ from the other main results (particularly in Figure 8 and Figure 9).

### G.1   LEARNING REWARDS ONLINE

In Figure 8 of the main text, we measure the gap in performance between pre-trained agents that have no knowledge of the reward function at the beginning of fine-tuning and agents whose reward predictor is initialized from a reward predictor learned on top of the unsupervised pre-training data (violating the URLB settings). Crucially, the agent during unsupervised PT can learn the reward predictor without affecting neither the model learning or the exploration process. To not affect the model, gradients are stopped between the reward predictor and the rest of the world model. To not affect exploration, the rewards used to train the agent's actor and critic remain the intrinsic rewards, for exploration.

### G.2   ZERO-SHOT ADAPTATION

Using agents that have access to a PT reward predictor, we explore the idea of zero-shot adaptation using MPC, which is trying to solve the URLB tasks using only planning and the pre-trained world model and reward predictor. In order to obtain good performance, this assumes that the model correctly learned the dynamics of the environment and explored rewarding transitions that are relevant to the downstream task, during pre-training. In Figure 9 of the main text, we compare the results of performing MPC in a zero-shot setting (ZS) with the performance of an MPC agent that is allowed 100k frames for fine-tuning (FT). As for the MPC method, we employ MPPI (Williams et al., 2015). Because these experiments are particularly expensive to run, we just them on the agents trained with the Plan2Explore URL approach.

We observe that the performance of zero-shot MPC is generally weak. While it overall performs better than the non-pre-trained model, simply applying MPC leveraging the pre-trained world model

and reward predictor trained on the pre-training stage data is not sufficient to guarantee satisfactory performance. The fact that exploiting the fine-tuning stage using the same MPC approach generally boosts performance demonstrates that the model has a major benefit from the FT stage. Still, the performance of MPC generally lacks behind the actor-critic performance, suggesting that, especially in a higher-dimensional action space such as the Quadruped one, amortizing the cost of planning with actor-critic seems crucial to achieve higher performance.

### G.3 LATENT DYNAMICS DISCREPANCY

Model misspecification is a useful measure to assess the uncertainty or inaccuracy of the model dynamics. It is computed as the difference between the dynamics predictions and the real environment dynamics. The metric helps build robust RL strategies, that take the dynamics uncertainty into account while searching for the optimal behavior (Talvitie, 2018). However, with pixel-based inputs the dynamics of the environment are observed through high-dimensional images. And this in-turn could hurt the metric evaluation, since the distances in pixel space can be misleading. In our approach, we use a model-based RL agent that learns the dynamics model in a compact latent space $\mathcal{Z}$.

Our novel metric, *Latent Dynamics Discrepancy* (LDD), quantifies the "misspecification" of the learned latent dynamics accordingly. The metric quantifies the distance between the predictions of the pre-trained model and the same model after fine-tuning on a downstream task. However, as the decoder of the world model gets updated during fine-tuning, the latent space mapping between model states $z$ and environment states $s$ might drift. For this reason, we freeze the agent's decoder weights, so that the model can only improve the posterior and the dynamics. This ensures that the mapping $\mathcal{Z} \rightarrow \mathcal{S}$ remains unchanged and allows to compare the dynamics model after fine-tuning with the one before fine-tuning. In order to measure the distance between the distribution output by the dynamics network, we chose the symmetrical Jensen-Shannon divergence:

$$\text{LDD} = \mathbb{E}_{(z_t, a_t)}\big[D_{\text{JS}}[p_{\text{FT}}(z_{t+1}|z_t, a_t)\|p_{\text{PT}}(z_{t+1}|z_t, a_t)]\big], \tag{3}$$

where the expectation is taken over the previous model states $z_t$ sampled from the fine-tuned posterior $q_{\text{FT}}(z_t)$, actions $a_{t-1}$ sampled from an oracle actor $\pi^*(a_t|z_t)$, so that we evaluate the metric on optimal trajectories, whose environment's state distribution corresponds to the stationary distribution induced by the actor $s_t \sim d^{\pi^*}(s_t)$. We used 30 trajectories per task in our evaluation.

We observe in our experiments that there exists a correlation between the metric and the performance ratio between a zero-shot model and a fine-tuned model (see Figure 10 in the main paper). The key observation is that major updates in the model dynamics during fine-tuning phase played an important role in improving the agent's performance, compared to the pre-trained model and zero-shot performance. Future research may attempt to reduce such dependency by either improving the model learning process, so that the pre-trained dynamics could have greater accuracy, or the data collection process, proposing URL methods that directly aid to reduce such uncertainty.

### G.4 UNSUPERVISED REWARDS AND PERFORMANCE

We further analyzed the correlation between the normalized performance of the different exploration agents and their intrinsic rewards for optimal trajectories obtained by an oracle agent. A strong negative correlation between the two factors should indicate that the agent is more interested in seeing the optimal trajectories when its performance is low on the task.

We observe that there is negative correlation between Plan2Explore (P2E), ICM, LBS's performance and their intrinsic rewards, while we found $\sim 0$ correlation for RND (see Table 1 in the main text). Out of the methods tested, LBS significantly demonstrated the correlation, as its p-value is $< 0.05$. This is likely one of the key factors for the high performance of the agent using LBS on the benchmark.

One possible explanation is that LBS searches for transitions of the environment that are difficult to predict for the dynamics, so the model likely learns those transitions more accurately, facilitating planning during the fine-tuning stage. Another potential explanation is that, given the high correlation between intrinsic and extrinsic rewards, the actor initialized by LBS performs better at the beginning of FT, speeding up adaptation.

## H  HYPERPARAMETERS

Most of the hyperparameters we used for world-model training are the same as in the original DreamerV2 work (Hafner et al., 2021). Specific details are as outlined here:

| Name | Value |
|---|---|
| **World Model** | |
| Batch size | 50 |
| Sequence length | 50 |
| Discrete latent state dimension | 32 |
| Discrete latent classes | 32 |
| GRU cell dimension | 200 |
| KL free nats | 1 |
| KL balancing | 0.8 |
| Adam learning rate | $3 \cdot 10^{-4}$ |
| Slow critic update interval | 100 |
| **Actor-Critic** | |
| Imagination horizon | 15 |
| $\gamma$ parameter | 0.99 |
| $\lambda$ parameter | 0.95 |
| Adam learning rate | $8 \cdot 10^{-5}$ |
| Actor entropy loss scale | $1 \cdot 10^{-4}$ |
| **Dyna-MPC** | |
| Iterations | 12 |
| Number of samples | 512 |
| Top-k | 64 |
| Mixture coefficient (Actor/CEM) | 0.05 |
| Min std (fixed) | 0.1 |
| Temperature | 0.5 |
| Momentum | 0.1 |
| Planning horizon | 5 |
| **Common** | |
| Environment frames/update | 10 |
| MLP number of layers | 4 |
| MLP number of units | 400 |
| Hidden layers dimension | 400 |
| Adam epsilon | $1 \cdot 10^{-5}$ |
| Weight decay | $1 \cdot 10^{-6}$ |
| Gradient clipping | 100 |

Table 4: World model, actor-critic, planner (Dyna-MPC) and common hyperparameters.

For the pure MPC-based experiments, we increased the number of MPPI samples from 512 to 1000, the number of top-k from 64 to 100, and the horizon from 5 to 15, to compensate for the absence of the actor network's samples and the critic's predictions in the return estimates.

