# OpenReview forum: "Unsupervised Model-based Pre-training for Data-efficient Control from Pixels"
_ICLR.cc/2023/Conference — Submitted to ICLR 2023_

### Official Review · Reviewer_xEiB · 2022-10-23

**Confidence:** 4
**Correctness:** 3
**Technical Novelty And Significance:** 2
**Empirical Novelty And Significance:** 4
**Recommendation:** 8

**Clarity, Quality, Novelty And Reproducibility:**

Clarity: the paper is really well-written, with the overall structure summarised concisely in Section 3.4

Novelty of the work is in a large scale evaluation; although the work is engineered out of the existing components, the reviewer thinks that it is important that the work shows the impact of architectural choices and therefore serves a good contribution.

Comments:
- In figure 2, it appears that while the proposed procedure wins over results from Laskin et al (2021) by a big margin,  the results still keep improving even after 2M frames pretraining; do the authors know what would happen beyond 2M frames, and if there is a point where no more pretraining iterations give improvement in fine-tuning? Does it stabilise, or does it lead to some form of overfitting degrading performance after certain point?
- Section 3.2: what is the impact of finetuning just actor, without fine-tuning the model?
- In Figure 6, is it possible to add the baseline of the proposed method ablation, where there is no pretraining phase, to match up with DrQ @100K? That would help complete the picture of the contribution of pretraining to the overall process.
-  Did the authors think of using DynaMPC in conjunction with the DrQ model? Is it possible (I guess not because if the world model requirement which is would make such modification model-based), and could it bring a fraction of the proposed benefits?
- In Algorithm 1, it looks like the world model $\phi$ is listed as the input but never explicitly referenced in the algorithm body; could the authors clarify whether this is correct and whether the explicit reference is needed?


**Strength And Weaknesses:**

Strengths:
- The paper is really well written: the narrative is shaped around choosing different components for pre-training and fine-tuning, with the structure of the overall approach neatly summarised in Section 3.4
- It contains compelling evidence of benefits of using proposed model-based reinforcement learning method over existing state-of-the-art model free technique
- The analysis includes necessary justification of the architecture: choice of the best possible pretraining model

Weaknesses:
- The model is engineered out of the existing components (apart from incremental, in a positive sense of this word, contribution of Dyna-MPC); this is not, however, a problem at all in my opinion as we need large-scale studies showing the impact of architectural choices, and this is a good contribution for the community and this very knowledge of how to improve architectural choices is a novel aspect of the paper

**Summary Of The Paper:**

The paper considers the task of unsupervised pretraining of the reinforcement learning (RL) agents.
It analyses and compares different design choices for pretraining and fine-tuning the pretrained components; it shows the improvement of using proposed model-based reinforcement learning over existing model-free technique, DrQ.

**Summary Of The Review:**

Based on the merits of the evaluation and experimental contribution, as well as the quality of writing, I recommend acceptance.

---

> ### Author Response · Authors · 2022-11-10
> **Response to Reviewer xEiB**
>
>
> We thank the reviewer for the useful feedback and for highlighting the quality of the work, experiments, and analysis.
>
> * The reviewer raised a question about whether longer pre-training could improve performance further. We ran additional experiments, which we present in the revised version of the paper's Appendix (Figure 14), and can also be found here: https://imgur.com/woCiOrn. These confirm that additional pre-training does improve performance further, especially until 4M pre-training steps. Thus, under the assumption that collecting data with unsupervised RL is "cheap" to do, our method scales well with additional data. We aim to provide a more extensive evaluation and investigation of this finding, adopting different URL strategies, in the final version of the paper.
> * Given that we ground on a latent space model, the actor predictions rely on the model inferred latent state. As a consequence, it is not possible to test the actor without the model. Results of a pre-trained model-free actor are likely similar to those of DrQ + X approaches.
> * We added the additional baseline(Dreamer@100k) in the revised version of the paper.
> * DrQ + Dyna-MPC could not be possible unless turning DrQ into a model-based approach. Other MPC-based approaches that we reported, such as LOOP [1] present a combination of model-free critic learning with model-based planning, but they do not allow learning in imagination, which is the main advantage of adopting Dyna-MPC in this setting. However, such a combination could still exploit a fair pre-trained model for online planning
> * We thank the reviewer for noticing we should have explained how the model is used in Algorithm 1. We revised the Algorithm in our updated manuscript to include that.
>
> We hope this answers the reviewer's questions. We also invite the reviewer to check out the code, that we just released in the Supplementary Material, as we believe this could be very useful to the community, to further re-use our approach.
>
> [1] Silkchi et al 2021. Learning Off-Policy with Online Planning.

---

> ### Author Response · Authors · 2022-11-24
> **Discussion**
>
> Dear reviewer,
>
> Could we please ask you to confirm whether our comments and revision resolved your concerns or whether there are any remaining issues?
>
> We would be happy to incorporate any additional feedback into our work or further clarify where necessary.
>
> We would also like to inform you that more comprehensive results about additional pre-training (up to 5M frames) have been uploaded in the revised manuscript.
>
> Thank you!

---

> > ### Comment · Reviewer_xEiB · 2022-11-24
> > **Re: discussion**
> >
> > Thanks for the question and for the reminder! It's all received, I am reading the discussions with other reviewers and the revision, and I will come back soon to respond to this question.

---

> > > ### Comment · Reviewer_xEiB · 2022-12-13
> > > **A note on the discussions**
> > >
> > > Dear authors,
> > >
> > > I'm just writing to send you a quick note on the results of our discussions and thank your hard work on the paper.
> > >
> > > I had a comprehensive look at the revision and other reviews, and while I maintain that this is a really good paper, we had a discussion, and the other reviewers had important points which I think are necessary to be addressed. These points, mainly, show the importance of showing  the trade-offs of the proposed pretraining algorithm on a range of tasks, as well as the limitations of URLB for benchmarking unsupervised pretraining:
> > > - 12HB said: *'First (W1), URLB is not a suitable setting to study unsupervised RL in full generality, as it solely involves locomotion in the DeepMind Control Suite and scopes out several lines of work evaluating unsupervised RL in the context of manipulation, navigation, and game-playing (Atari). Second (W2), the first detailed contribution of "show[ing] that world models pre-trained with data collected using unsupervised RL can facilitate adaptation for future tasks" (abstract) is already well-established knowledge contributed by earlier Plan2Explore/Dreamer works on DMCS tasks, as well as e.g. LEXA for manipulation tasks.'* While the authors have done their best to amend the text, it might require more work to fully address it. I agree that showing the limitations of URLB as a benchmark, as well as highlighting whether or not proposed unsupervised pretraining could help improve the performance on manipulation, navigation, pixel-based game-playing.

---

### Official Review · Reviewer_kuHx · 2022-10-24

**Confidence:** 4
**Correctness:** 2
**Technical Novelty And Significance:** 2
**Empirical Novelty And Significance:** 2
**Recommendation:** 3

**Clarity, Quality, Novelty And Reproducibility:**

The paper is unclear to read in some parts, is fairly high quality, lacks quite a bit on novelty, and is decently reproducible.

**Strength And Weaknesses:**

Strengths

- Important and relevant problem setup
- Fairly thorough experiments

Weakness

- Lack of a coherent story
- Unclear primary contributions
- Somewhat hard to read through

**Summary Of The Paper:**

The paper studies the problem of reward free learning, where the setup involves a pre-training stage (when an intrinsic reward is used based on different prior methods) and a fine-tuning stage (when a particular extrinsic reward is provided). The claim is that model-based methods perform better than model-free methods in such a setup. The authors perform certain ablations where different components of the model-based method (in this case the actor, critic, and dynamics model of the Dreamer algorithm) are shown to affect fine-tuning performance differently. Finally, the authors deploy a MPC style planning algorithm instead of using the learnt actor when fine-tuning to new rewards.

**Summary Of The Review:**

The paper lacks a coherent story and it’s hard to parse the clear contributions it makes. The Dyna-MPC algorithm as well as the ablations are not the main contributions in my opinion. As far as the main claim goes, I am not sure why this is surprising, i.e. shouldn’t any model-based method be expected to perform better than model-free method when the reward function is changed? I could not understand why Figure 2 has a flat line for the Drq@100k and Dreamer@100k cases. What does 100k refer to here? Can you provide some commentary on why, in the setup you consider, model-based methods should outperform model-free ones by such a huge margin? I think this is crucial to the reader’s understanding of the contributions and motivation of the paper.

---

> ### Author Response · Authors · 2022-11-10
> **Response to Reviewer kuHx**
>
> We would like to thank the reviewer for the feedback and for appreciating the importance of the problem and our thorough experimentation.
>
> > The paper lacks a coherent story and it’s hard to parse the clear contributions it makes. The Dyna-MPC algorithm as well as the ablations are not the main contributions in my opinion.
>
> We understand the paper may be complex, given the several studies we included and we would like to understand better what the reviewer finds difficult to parse, in order to improve the current structure/story-telling. We note that other reviewers found our paper "very clear", "well structured" and "really well-written". Currently, we tried to indicate what is the main scope of the paper as follows:
> * In the Abstract, we highlight that the main idea behind the paper is "the design of an unsupervised RL strategy for data-efficient visual control", which is missing in the literature around the URLB benchmark.
> * In the Introduction, we highlight that our new planner and the empirical ablations should not be considered the main contributions of the work, but rather building blocks towards the development of an effective strategy for this difficult setting, when saying: "This work does not propose a novel complex method. Rather, we study the interplay of various existing components and propose a novel final solution that outperforms existing state of the art on URLB by a staggering margin"
> * In Section 3.4, we restate our main objective, which is "exploring several design choices to establish the most adequate approach to tackle the URL benchmark" and we summarize our findings.
>
> > As far as the main claim goes, I am not sure why this is surprising, i.e. shouldn’t any model-based method be expected to perform better than model-free method when the reward function is changed?
>
> We agree that model-based methods are expected to perform better than model-free methods when the dynamics of the environment stays the same and reward changes. However, our findings demonstrate several other level of improvements beyond simply using a model-based approach.
> As the reviewer's statement highlights, it is important to report model-based experiments on URLB as they are currently absent from the literature. However, our work is not limited to that. As we progress through the paper we show that the empirical performance can significantly change depending on some design choices, such as the exploration approach used (e.g. competence-based methods tend to perform worse than data/knowledge-based)  or which components are used during the adaptation (fine-tuning) stage. In this context, Dyna-MPC is a contribution that provides an additional significant improvement in performance, once again proving that a good pre-trained model can strongly support fast adaptation.
>
> > I could not understand why Figure 2 has a flat line for the Drq@100k and Dreamer@100k cases. What does 100k refer to here?
>
> The "@100k" means the supervised agent is trained from scratch for 100k steps. It is flat because this agent is not using the pretraining environment. To clarify this, we added the following to the Figure caption in the updated manuscript (in orange text):
> > We also report Dreamer@100k and DrQ@100k results, which are obtained in 100k FT steps with no PT.
>
> We also made the lines different (dashed) to differentiate from the remaining approaches.

---

> > ### Author Response · Authors · 2022-11-10
> > **Part 2 of response**
> >
> > > Can you provide some commentary on why, in the setup you consider, model-based methods should outperform model-free ones by such a huge margin? I think this is crucial to the reader’s understanding of the contributions and motivation of the paper.
> >
> > When learned end-to-end, the actor and critic functions for the pseudo-reward may be fundamentally different from the task-reward. In such case, starting from a pre-trained actor-critic can lead to very little improvement or can even be worse than training from scratch. As suggested by the reviewer, we updated the paper including the following statement in the Preliminaries section to improve the reader's understanding of this issue (in orange in the updated manuscript):
> > > In this setting, the performance of unsupervised model-free RL [1] were shown to be insufficient as reported in [2]. We believe the key reason for this is that model-free RL algorithms can exploit only a little part of the information obtained with self-supervised interaction, as they rely uniquely on actor and critic's predictions.
> > >
> > In URLB, the underlying dynamics is assumed to stay the same between unsupervised pre-training and task-driven fine-tuning. For this reason, if we would perfectly learn the dynamic of the environment during the pre-training phase, the only thing left to learn about the downstream task is the reward function, and it can be done quickly and efficiently, as we demonstrate.
> >
> > We understand there is a strong assumption about the environment dynamics, and we discuss this in Section 4.2 (RWRL experiments), as well as testing performance in environments that share similar dynamics, but present physical and sensorial perturbations that make the model less valid during fine-tuning. Here, we found that pre-training is still beneficial, unless the perturbations become too strong.
> >
> > We hope this clarifies our contributions and we look forward to the reviewer's feedback, in order to further improve the current structure of the paper. We also invite the reviewer to check out the code, that we just released in the Supplementary Material, as we believe this could be very useful to the community, to further re-use our approach.
> >
> > [1] Yarats et al 2021 Mastering Visual Continuous Control: Improved Data-Augmented Reinforcement Learning
> >
> > [2] Laskin et al 2021. URLB: Unsupervised Reinforcement Learning Benchmark

---

> ### Author Response · Authors · 2022-11-24
> **Discussion**
>
> Dear reviewer,
>
> Could we please ask you to confirm whether our comments and revision resolved your concerns or whether any remaining issues motivate your current score?
>
> We would be happy to incorporate any additional feedback into our work or further clarify where necessary.
>
> Thank you!

---

### Official Review · Reviewer_12hB · 2022-10-24

**Confidence:** 4
**Correctness:** 3
**Technical Novelty And Significance:** 2
**Empirical Novelty And Significance:** 2
**Recommendation:** 5

**Clarity, Quality, Novelty And Reproducibility:**

Clarity

The writing was very clear and well-structured.

Quality

The execution of the work appears sound. The empirical results are impressive.

Novelty

As the authors admit, the work does not propose a novel method, but rather demonstrates the outsized benefit of applying existing methods to a particular problem setting. As noted above, this benefit is itself not particularly novel, as the point was already previously made in the Plan2Explore work. However, this work does provide quite a few minor novelties over Plan2Explore, including swapping out Dreamer for DreamerV2, assessing using a benchmark not available to Plan2Explore, assessing using a variety of self-supervised reward mechanisms during pre-training, investigating the transfer of various components, the Dyna-MPC planning mechanism, and robustness results on RWRL.

Reproducibility

I'm satisfied with the level of detail provided for reproducibility purposes.

**Strength And Weaknesses:**

Strengths

The results are particularly strong. The progression of experiments building towards the proposed method is well structured and informative. The set of methods, comparisons, and ablations is extensive.

Weaknesses

The gains provided by model-based learning are probably significantly inflated due to the simplicity of the three URLB environments, which enables in-imagination training and planning to actually work. In contrast, in Atari, which has far more visual variation per game despite arguably simpler dynamics, model-based methods have still yet to come close to model-free methods, except when the allowed amount of environment interaction or compute is limited. Thus, while impressive, the improvements coming from this work should be taken with a grain of salt, and might not hold for environments with more realistic amounts of variation and complexity.

The application of model-based learning for unsupervised pre-training for RL has already been known to be effective (e.g. Plan2Explore). I'm not sure why the URLB authors did not benchmark model-based baselines like Plan2Explore, but regardless, the improvements over the DrQ-based runs is expected given Plan2Explore, especially given the environments in Plan2Explore and URLB are both sourced from the DeepMind Control Suite. Regrettably, it seems that the authors formulated their main hypothesis without realizing the above. I recommend rephrasing the contributions in the abstract and intro to convey that the results in this work corroborate and extend upon the findings first provided by Plan2Explore.



**Summary Of The Paper:**

The authors study model-based methods for tackling the unsupervised reinforcement learning benchmark (URLB). First, the authors show that substituting DrQ (a model-free agent) with DreamerV2 results in increased performance and scaling on URLB. Second, the authors investigate the transfer of specific components (model, actor, critic) from pre-training to fine-tuning, finding that transferring model and actor is beneficial but additionally transferring critic compromises performance. Third, the authors propose Dyna-MPC, a planning procedure that builds upon model predictive path integral (MPPI) control, using the actor as an additional source of sampled trajectories, and the critic and model to score trajectories. This is shown to further improve fine-tuning. These improvements altogether result in dramatic improvement on URLB. Finally, the authors assess transfer under distribution shifts via the real-world reinforcement learning benchmark (RWRL), and find that their proposed model without Dyna-MPC performs best.

**Summary Of The Review:**

This work is rather tricky to assess. While its proposed methods do provide dramatically improved empirical results on an existing benchmark, as argued above there are important limitations to both the novelty and significance in this improvement. However, the thoroughness with which the empirical evaluation was conducted is admirable, and there are a couple of genuinely new insights provided on the application of model-based learning to unsupervised RL. I currently lean towards rejection, but look forward to discussion with the authors and other members of the reviewing team to inform my final recommendation.

---

> ### Author Response · Authors · 2022-11-10
> **Response to Reviewer 12hB**
>
>
> We thank the reviewer for the constructive feedback and for positively highlighting that the paper is well-structured and our evaluation strong and extensive.
>
> * The reviewer highlights that our method works very well in URLB, which is a setting where the dynamics between PT and FT stages stays nearly the same, but its performance may be less strong in cases where the dynamics differs between the two. We believe that the fact we nearly solve the URLB benchmark is an important contribution of our work: it demonstrates the need of new benchmarks for helping the URL community to progress in a more challenging setup. The Atari benchmark may represent an example in that respect, but considering that every game represents a domain on its own, given the large discrepancies between their tasks and dynamics, it would be hard to assess generalization.
> In this respect, our experiments on RWRL aim to see whether some of the advantages of our method hold when the dynamics between PT and FT differs. Our results are encouraging as there is still a significant improvement when the visual variations and noise in the observations are not too strong. We also highlight the importance, for future work, to develop models that can generalize better to such disturbances in the Limitations paragraph of the Discussion:
>
> > Another issue we encountered, on the RWRL benchmark, is that if the environment introduces too intense perturbations during adaptation, relying on the predictions of the adopted world model becomes problematic, to the extent that exploiting a planner is not useful anymore. Developing more resilient models that can be trained in an unsupervised fashion and used for data-efficient planning, even in presence of complex perturbations, will be the focus of future studies.
>
> * According to the reviewer's suggestion, we rephrased our hypothesis in the Introduction as "Inspired by previous work on exploration [1]" (in orange in the updated manuscript). With respect to this previous work (Plan2Explore), we introduced several important design choices and additions (i.e. Dyna-MPC) that strongly improved performance, as shown in the paper and in the Appendix. To further highlight the improvement in performance that we obtained, we summarize the performance in a table:
>
> | Approach | Performance (overall avg) | New configuration |
> | -------- | -------- | -------|
> | Plan2Explore - PT model     |   70.34 $\pm$ 12.52   |  |
> | Plan2Explore - PT model, actor, critic     |  76.59 $\pm$ 20.81 |
> | Plan2Explore - PT model, actor     |   81.85 $\pm$ 16.28   | X |
> | Plan2Explore - FT     |   83.08 +- 12.05   | X |
> | Plan2Explore - FT + Dyna-MPC     |  88.86 +- 10.57    | X |
> | LBS - FT + Dyna-MPC     |   93.59 +- 6.50   | X |
>
>
> Considering the standard fine-tuning strategies "PT model" and "PT model and actor-critic" as the original Plan2Explore configurations, we see that the new configurations we developed managed to improve performance by:
> * +5.3%, when excluding the critic
> * +1.2%, when not initializing the actor for Jaco (FT)
> * +5.8%, using our new hybrid planner Dyna-MPC
> * +4.7%, when changing the unsupervised RL strategy
>
> For a total of **+17%** over the original method, which we find a significant improvement and should bring useful findings to the community.
>
> We hope this revision helps informing the reviewer's recommendation. We also invite the reviewer to check out the code, that we just released in the Supplementary Material, as we believe this could be very useful to the community, to further re-use our approach.
>
> [1] Sekar et al 2020. Planning to Explore via Self-Supervised World Models.

---

> > ### Comment · Reviewer_12hB · 2022-12-01
> > **Review update**
> >
> > These are my updated reviewing comments after the author response and after a meeting with the reviewing team.
> >
> > Assessment of the paper
> > - The paper's main strength (S1) lies in its execution and the contribution of the empirical bag of tricks that elevate model-based unsupervised pre-training for RL, as succinctly summarized in your above table.
> > - However, the paper has several weaknesses. First (W1), URLB is not a suitable setting to study unsupervised RL in full generality, as it solely involves locomotion in the DeepMind Control Suite and scopes out several lines of work evaluating unsupervised RL in the context of manipulation, navigation, and game-playing (Atari). Second (W2), the first detailed contribution of "show[ing] that world models pre-trained with data collected using unsupervised RL can facilitate adaptation for future tasks" (abstract) is already well-established knowledge contributed by earlier Plan2Explore/Dreamer works on DMCS tasks, as well as e.g. LEXA for manipulation tasks.
> >
> > Recommendations
> > - Any language supporting the claim underlying W2 should be removed. The authors have already made a token modification in the introduction to this end, but significant examples of this still remain, e.g. in the abstract as I quoted above.
> > - W1 can be addressed in two ways. The authors can re-scope the contribution of the paper as showing how URLB is unsuitable as evaluation for unsupervised RL. Or, the authors can expand the scope of their study and deploy their well-tuned model-based configurations to other unsupervised RL domains. If their model-based method dominates, great! And if the authors find a setting where model-based methods fail to show similar gains, even better: a nuanced demarcation of the trade-off between different unsupervised RL methods with respect to key properties of different application domains would be a far more significant contribution. In either case, the impact of the work will be greatly amplified.
> > - Multiple other reviewers mentioned that the core message of the work was not clear. I agree to the extent that W1 and W2 muddy S1, but if the other reviewers have other comments on this front, I strongly encourage the authors to incorporate their feedback.
> >
> > Score
> > - My score remains a 5.

---

> > > ### Author Response · Authors · 2022-12-06
> > > **Additional comments 1/2**
> > >
> > >
> > > We would like to thank the reviewer for the additional feedback and for participating in the discussion. Here find our comments.
> > >
> > > > First (W1), URLB is not a suitable setting to study unsupervised RL in full generality, as it solely involves locomotion in the DeepMind Control Suite and scopes out several lines of work evaluating unsupervised RL in the context of manipulation, navigation, and game-playing (Atari).
> > >
> > > URLB aims to test a limited set of the agent's capabilities, namely exploration and fast adaptation in DMC suite environments where the dynamics remains unchanged between pre-training and fine-tuning. We agree with the reviewer that URLB has its limitations, which are also discussed in the original paper. However, as stated by the URLB authors, "existing algorithms are unable to solve the benchmark meaning there is substantial room for improvement on the URLB tasks before moving on to even more challenging ones" [URLB].
> > >
> > > Our work nearly solves the URLB benchmark, indeed **showing that URLB is not suitable for future research on URL**. We see this as an important contribution of our work rather than a weakness.  Without our work, the community would still focus on this benchmark. On the other hand, our paper is likely to spur the development of a new benchmark, and the provided baselines can help selecting challenging tasks. In summary, the influence that this work can have on the URL community is large and important.
> > >
> > > Following the reviewer's suggestion that it should be clear that "URLB is unsuitable as evaluation for unsupervised RL", we added the following paragraphs to our paper:
> > >
> > > * (Introduction - Contributions paragraph, after the bullet list) "Our results, showing that we are able to nearly solve the URLB benchmark from pixel inputs, also demonstrate that new more complex benchmarks will be necessary in order to foster further advancements in unsupervised RL research."
> > > * (Discussion - before the Limitations paragraph) "Finally, one important finding of our work is that the URLB benchmark, which has been recently popularized in the URL community, may be insufficient to further advance current methods. In the future, we aim to consider more complex benchmarks, e.g. where the environment appearance distribution changes as the agent proceeds [ATARI,PROCGEN], to investigate to what extent our unsupervised model-based pre-training strategy still brings its benefits."
> > >
> > > [URLB] URLB: Unsupervised Reinforcement Learning Benchmark, Laskin et al, 2021
> > >
> > > [ATARI] The Arcade Learning Environment: An Evaluation Platform for General Agents, Bellemare et al, 2013
> > >
> > > [PROCGEN] Leveraging Procedural Generation to Benchmark Reinforcement Learning, Cobbe et al, 2020

---

> > > > ### Author Response · Authors · 2022-12-06
> > > > **Additional comments 2/2**
> > > >
> > > > > Second (W2), the first detailed contribution of "show[ing] that world models pre-trained with data collected using unsupervised RL can facilitate adaptation for future tasks" (abstract) is already well-established knowledge contributed by earlier Plan2Explore/Dreamer works on DMCS tasks, as well as e.g. LEXA for manipulation tasks.
> > > >
> > > > To the best of our understanding, Plan2Explore is limited in showing that unsupervised RL combined with world models can facilitate adaptation in that it shows that only Disagreement [Disag] on latent space is effective. In particular, the 'Curiosity' baseline presented in the Plan2Explore paper shows that an ICM-like [ICM] intrinsic reward fails to provide good exploration when combined with a world model.
> > > >
> > > > Differently, in our work, we show that, with careful implementation, several URL strategies, including ICM, are effective in facilitating adaptation. As a matter of fact, with our approach, ICM, LBS, APT, and RND all perform better than Plan2Explore in terms of IQM, median, mean, and optimality gap (see Figure 5 of the paper).
> > > >
> > > > For this reason, we believe that our claim of showing that "world models together with unsupervised RL facilitate adaptation" holds to some extent. However, as the reviewer suggested, the way we presented it may be misleading. To resolve the issue, we rephrased the contribution as:
> > > >
> > > > * (Abstract) "we show **how several unsupervised RL approaches** can be used to pre-train world models and facilitate adaptation for future tasks"
> > > > * (Introduction - Contributions paragraph) "we **show how multiple unsupervised RL approaches combined with world models** can enable data-efficient visual control **in the context of URLB**"
> > > >
> > > > We again thank the reviewer for participating in the discussion and we hope our comments clarify our contributions and resolve the issues raised.
> > > >
> > > > [Disag] Self-Supervised Exploration via Disagreement, Pathak et al, 2019
> > > >
> > > > [ICM] Curiosity-driven Exploration by Self-supervised Prediction, Pathak et al, 2017

---

> ### Author Response · Authors · 2022-11-24
> **Discussion**
>
> Dear reviewer,
>
> Could we please ask you to confirm whether our comments and revision resolved your concerns or whether any remaining issues motivate your current score?
>
> We would be happy to incorporate any additional feedback into our work or further clarify where necessary.
>
> Thank you!

---

### Official Review · Reviewer_zaP8 · 2022-10-25

**Confidence:** 5
**Correctness:** 4
**Technical Novelty And Significance:** 2
**Empirical Novelty And Significance:** 2
**Recommendation:** 6

**Clarity, Quality, Novelty And Reproducibility:**

The paper is clearly written and easy to follow. The method itself is built upon existing methods in the literature, but novelty itself is not a limitation given the strong empirical performance.

**Strength And Weaknesses:**

Strength
* This work offers a large-scale benchmarks for different URL techniques and compare the usefulness of different pre-trained components.
* The developed method has strong empirical performance.

Weaknesses
* The main weakness is novelty since most components in this work come from existing works.
* The paper pinpoints two decisions to make to apply the proposed framework for tasks outside URLB in Section 3.4. Within URLB, these decisions are made given empirical benchmarking results. But more insights or explanations of when and why different techniques work in different settings, and how much alignmenent of the pretraining and downstream task the proposed mechanism can handle, will further strengthen the contribution of this work.

**Summary Of The Paper:**

The method studies the generalization capabilities of unsupervised RL empirically, and develops a hybrid planner with strong asymptotic performance and high sample efficiency in standard benchmarks.

**Summary Of The Review:**

The empirical findings from the work are insightful for the community, and the proposed method stemming from these findings achieves strong performance in standard benchmarks. Both are valuable contributions to the community.

---

> ### Author Response · Authors · 2022-11-10
> **Response to Reviewer zaP8**
>
>
> We thank the reviewer for the useful feedback and for highlighting that our empirical finding and resulting algorithms are valuable contributions to the community despite the limited novelty.
>
> As the reviewer noted, in Section 3.4, we summarize our finding along with pinpointing some design choices that we found crucial in improving performance on the URLB benchmark, which are (a) whether fine-tuning the PT actor is meaningful for the downstream task or it's better to re-learn it from scratch, (b) what is the best URL strategy to collect data.
>
> To provide further insights to future readers we added the following to Section 3.4 (orange text in the updated manuscript):
> > Both decisions strongly depend on the target domain/task and so it is difficult to assess their implications beforehand, as this will depend on how well the unsupervised behavior of the PT actor correlates with the task to solve. However, adopting unsupervised strategies that specifically focus on interacting with interesting elements of the environment, e.g. objects, or that quickly explore large areas of the environment at the beginning of fine-tuning may help exploring and revisiting crucial states of the environment more easily[1].
> [1] Parisi et al 2021. Interesting Object, Curious Agent: Learning Task-Agnostic Exploration.
>
> In addition, in the Limitations paragraph of the Conclusion, we mentioned that improving current competence-based approaches could lead to better initializations of the actor, thanks to the multiple skills available. Namely:
> > In the Jaco reaching tasks, we found that a bad initialization of the pre-trained actor can actually harm the agent’s performance. While competence-based approaches should address this limitation, by learning a variety of skill behaviors, their performance on the other domains has been subpar. Future work should aim to find a more general approach to pre-train behavior for fast adaptation or improve the exploration capabilities of competence-based approaches.
>
> We hope this revision improves our paper. We also invite the reviewer to check out the code, that we just released in the Supplementary Material, as we believe this could be very useful to the community, to further re-use our approach.

---

> ### Author Response · Authors · 2022-11-24
> **Discussion**
>
> Dear reviewer,
>
> Could we please ask you to confirm whether our comments and revision resolved your concerns or whether any remaining issues motivate your current score?
>
> We would be happy to incorporate any additional feedback into our work or further clarify where necessary.
>
> Thank you!

---

> > ### Comment · Reviewer_zaP8 · 2022-11-30
> > **Thank you for your response**
> >
> > Dear authors,
> >
> > Thank you for your responses and edits to the manuscript. My score keeps the same since I think there are technical contributions in this work and it sets up a good baseline for future research in the area. I also agree with reviewer 12hB's comments on Atari and I believe further investigation in these domains, with a larger gap between model-based and model-free methods, will provide further insights.

---

### Author Response · Authors · 2022-11-24
**General response**

We would like to thank all reviewers for the time they spent reviewing our work and providing constructive feedback.

We are glad that all reviewers found our results significant and our experimental analysis thorough. For these reasons, we believe our large-scale study brings important information to the community and further highlights the need for new benchmarks since we nearly solved the URLB one. Furthermore, if published, our method could become the main baseline in this research area where, otherwise, the best results would be the ones from the URLB paper, obtaining ~60% lower performance than ours. Finally, we expect these results to be particularly influential in the next developments of the URL community and help focus new research on solving more challenging tasks.

We addressed the concerns raised by the reviewers in the individual responses and we revised our manuscript according to the suggestions provided. Here we summarize the main changes:
* (**all**) we released the code in the Supplementary Material
* (**zaP8**) we provided further insights on how to deal with some of the design choices we studied, in Section 3.4
* (**12hB**) we rephrased our hypothesis in the Introduction
* (**12hB**) we presented an overview table quantifying how we significantly improved performance over previous work (this could be included as part of the final version of the paper, if the reviewers find it useful)
* (**kuHx**)  we clarified some of the baselines presented in the plots
* (**kuHx**) we added a discussion on why model-free approaches struggle in the URLB setting, in the Preliminaries section
* (**xEiB**) we presented additional experiments in the Appendix with longer pre-training (up to 5M frames), to answer one of the reviewer's questions
* (**xEiB**) we improved our plots and algorithms presentation according to the reviewer's suggestions

We hope our revision addresses the raised concerns and we would be happy to incorporate additional feedback or respond to further questions.

---

### Decision · Program_Chairs · 2023-01-20

**Decision:**

Reject

**Justification For Why Not Higher Score:**

The reviewers raised concerns regarding the generality of the claims and also the novelty of the claims. The AC considers these concerns as valid and cannot recommend acceptance.

**Justification For Why Not Lower Score:**

N/A

**Metareview: Summary, Strengths And Weaknesses:**

The paper performs an extensive study of model-based approaches for the URLB benchmark. The findings show that replacing a model-free approach with Dreamer V2 improves the performance. The paper performs further analysis by using pre-training or fine-tuning for the model, actor, and critic. The findings of the experiments leads to an approach that provides significant gains on URLB. The paper also demonstrates results on Real-Word RL benchmark to test generalization to noisy environments.

Strengths:
- The experiments are very well structured and thorough.
- The paper is also well written and clearly demonstrates the progression of the experiments towards building the proposed approach.
- The paper shows a massive improvement on the benchmark.

Weaknesses:
- Some of the findings of the paper (e.g., model-based methods outperforming model-free methods when the dynamics of the environment stays the same and reward changes) are expected.
- The results are limited to URLB, which is simplistic compared to more sophisticated benchmarks used by other unsupervised RL works. So, the findings will probably not generalize.


**Summary Of Ac-Reviewer Meeting:**

The paper received divergent ratings (3,5,6,8). So, the area chair and the reviewers discussed the paper extensively in a virtual meeting. All of the reviewers appreciated the thoroughness of the experiments and the provided gain on the benchmark. However, there are two major concerns. First, the benchmark is simplistic in today’s standards compared to several lines of work evaluating unsupervised RL in more complex scenarios. Hence, the findings will be limited to this benchmark. Second, some of the main claims of the paper on the effect of pre-training on unsupervised RL is well established by previous work (mentioned in the reviews).

Due to these issues, all reviewers except zaP8 voted for rejection. The AC agrees with the concerns and follows the recommendation of the majority and recommends rejection.